# Detection of Volatile Compounds Emitted by Bacteria in Wounds Using Gas Sensors

**DOI:** 10.3390/s19071523

**Published:** 2019-03-28

**Authors:** Carlos Salinas Alvarez, Daniel Sierra-Sosa, Begonya Garcia-Zapirain, Deborah Yoder-Himes, Adel Elmaghraby

**Affiliations:** 1eVIDA Research Group, University of Deusto, 48007 Bilbao, Spain; mbgarciazapi@deusto.es; 2Department of Computer Engineering and Computer Science (CECS), University of Louisville, Louisville, KY 40292, USA; d.sierrasosa@louisville.edu (D.S.-S.); adel@louisville.edu (A.E.); 3Department of Biology, University of Louisville, Louisville, KY 40292, USA; deborah.yoder-himes@louisville.edu

**Keywords:** gas sensors, *Pseudomonas*, Raspberry Pi, Arduino, wound infection

## Abstract

In this paper we analyze an experiment for the use of low-cost gas sensors intended to detect bacteria in wounds using a non-intrusive technique. Seven different genera/species of microbes tend to be present in most wound infections. Detection of these bacteria usually requires sample and laboratory testing which is costly, inconvenient and time-consuming. The validation processes for these sensors with nineteen types of microbes (1 *Candida*, 2 *Enterococcus*, 6 *Staphylococcus*, 1 *Aeromonas*, 1 *Micrococcus*, 2 *E. coli* and 6 *Pseudomonas*) are presented here, in which four sensors were evaluated: TGS-826 used for ammonia and amines, MQ-3 used for alcohol detection, MQ-135 for CO_2_ and MQ-138 for acetone detection. Validation was undertaken by studying the behavior of the sensors at different distances and gas concentrations. Preliminary results with liquid cultures of 10^8^ CFU/mL and solid cultures of 10^8^ CFU/cm^2^ of the 6 *Pseudomonas aeruginosa* strains revealed that the four gas sensors showed a response at a height of 5 mm. The ammonia detection response of the TGS-826 to *Pseudomonas* showed the highest responses for the experimental samples over the background signals, with a difference between the values of up to 60 units in the solid samples and the most consistent and constant values. This could suggest that this sensor is a good detector of *Pseudomonas aeruginosa*, and the recording made of its values could be indicative of the detection of this species. All the species revealed similar CO_2_ emission and a high response rate with acetone for *Micrococcus*, *Aeromonas* and *Staphylococcus*.

## 1. Introduction

Wound infections are a worldwide global health threat that compromises patient recovery. To solve this problem, we have previously studied chronic wounds such as pressure ulcers using the image segmentation technique [1] and classification via a 3D convolutional neural network [2,3]. Other studies [4] have also focused on the use of color image processing to detect infection in wounds.

One particular bacterial species, *Pseudomonas aeruginosa*, is a major causative agent in wound infections [5], constituting one of the main causes of infection in immunosuppressed hospitalized patients [6]. Another one of the principal causes of these infections is the bad smell that some wounds emit [7] due to dead organic matter or decomposed tissue. Putrescine and cadaverine—produced in decaying tissues—play an active role in producing putrefaction odors [8].

Proving that a wound is infected currently requires laboratory analysis, which is costly and time-consuming for the patient. Usually, the cost of a pressure ulcer treatment ranges from 15,000 € to 17,000 € in Europe [9] and can lead to an annual cost of $1.3 billion in United States [10]. In addition, access to laboratories capable of such testing is sometimes limited, especially in developing nations. All this can delay the appropriate treatment causing pain in the patient and increasing the risk of other infections.

Therefore, there is a need to develop faster and easier methods for detecting infections in wounds so that specialist laboratories would not be needed. Doctors could thus give more rapid and accurate diagnosis, which would accelerate the healing process and patients’ quality of life.

The use of health sensors has increased during the last few years with the development of sensors to monitor parameters such as snoring episodes [11], glucose [12] and detecting vital signs such as blood pressure [13].

Several studies [14,15,16,17,18] have demonstrated the emission of certain volatile gases by a major skin and soft tissue pathogen, *P. aeruginosa*. The main volatile gases emitted by *P. aeruginosa* are ammonia and amines; acetone and its variants; alcohols such as ethanol; and various gases resulting from the anaerobic decomposition of the tissue by this bacterium including putrescine and cadaverine.

For this study, we propose using low-cost gas sensors to detect both infection and the state of decomposition in wounds from the concentrations of putrescine and cadaverine [19]. We also tested the detection of CO_2_, ammonia and amines including putrescine and cadaverine, acetone, and alcohol to determine whether detection of these compounds might also be correlated with concentrations of *Pseudomonas aeruginosa*. Using sensors to monitor volatile compounds produced during skin infections would provide real-time data collection, as well as being a non-intrusive technique for rapid and early diagnosis.

This paper presents the results associated with the calibration and validation process of gas sensors and the testing process with bacteria commonly found in wound infections. The results provided constitute a fundamental first step in the construction of a non-invasive device based on Arduino hardware and on the selected sensors for detection of *P. aeruginosa* in situ.

## 2. Methods

A wide variety of sensors, usually designed to operate with high concentrations, have been developed to detect gases. Nonetheless, when properly calibrated, it is possible to detect lower concentrations with low fluctuations in their signals. In this section we present the criteria for the sensors selection, the validation process carried out on each sensor with ammonia, alcohol, CO_2_ and acetone and the system and experiment design with the electronic connections. The measurements are performed under controlled environmental conditions with room temperature between 20 °C and 22 °C.

### 2.1. Sensors Selection

The gas sensors which could be chosen to detect the gases emitted by nineteen different bacteria must meet certain criteria. Following some parameters evaluated for the electronic nose presented in [20,21], we define the following inclusion and exclusion criteria (Table 1 and Table 2).

For each type of gas, the technical characteristics between different sensors on the market were analyzed according to the inclusion and exclusion criteria. The sensors analyzed were MQ-137 and TGS-826 for ammonia detection, MQ-303A, MQ-3 and AL600 for alcohol, MQ-135, MG811 and SPRINTIR for CO_2_ and MQ-138 for acetone. The sensors selected and their technical characteristics are shown in Table 3.

The sensors presented in Table 3 provide an appropriate detection range and accuracy and operate on a 5 V supply which is suitable for use with an Arduino board. The detailed experimental design is presented in Appendix A, where the selected sensors connections are described.

### 2.2. Experiment Design

Six clinical strains of *P. aeruginosa* were retrieved from a −80 °C freezer and single colony isolations were used on Luria Broth Lennox formulation (LB) agar to obtain single colonies on LB solid agar. Each strain was grown in 5 mL LB broth overnight at 37 °C with shaking. Cultures were diluted as indicated into 4 mL of LB liquid in 60 mm sterile petri plates on the bench top for liquid culture experiments. For solid agar experiments, 100 μL of overnight culture of each strain was spread on the surface of 100 mm petri plates containing LB agar. Solid agar cultures were incubated at 37 °C overnight for ~18 hours prior to measuring. For quantification of bacterial concentrations liquid cultures were serially diluted and plated while 1 cm^2^ of solid agar cultures were scraped and re-suspended in 1 mL LB broth prior to serially dilution and plating. The six *P. aeruginosa* strain descriptions are summarized in Table 4.

Due to the fact that there are numerous types of bacteria living together in an infected wound, the gases emitted by 19 microbes frequently found in wounds were also analyzed. Strains of the following microbes were used:***Escherichia coli*:** this bacterium is among the main causes of all intra-hospital infections. It can cause skin diseases such as necrotizing fasciitis [22], wound infections (including those resulting from surgery) and infections in pressure ulcers [23].***Aeromonas hydrophila*:** a contributor to opportunistic wound infections and diabetic foot infections [24], in addition to showing clinical manifestations in the skin such as cellulitis, abscesses, or grangrenosis ecthymes [25].***Micrococcus luteus*:** an agent that helps to potentiate skin infection caused by other bacteria such as Staphylococcus aureus [26].***Enteroccocus faecalis*:** contributes to wound infections [27].***Candida*:** a contributor to skin candidiasis infections by fungi penetrating into the skin [28].

Quantitation and preparation of these microbial species were conducted as described above for *P. aeruginosa* strains in solid cultures.

### 2.3. Validation Process

In order to perform a systematic measurement, the signal should be stable, meaning that the sampling rate has to be defined. The concentration of the gas and the distance at which the samples are measured with the sensors are also important parameters that need to be studied and validated to see at which distances and concentrations the sensors lose their sensitivity.

The work flow carried out in each experiment is shown in Figure 1. The operational sample rate for each sensor was determined in the first step, and the effect of distance and gas concentration level were then assessed. In the next subsection each step of the work flow is described.

#### 2.3.1. Step 1: Operational Sample Rate

The sample rates of the sensors should be defined in order to record a stable signal. In order to achieve this, the read values that are recorded on the sensors should correspond to an average of several values taken in a very short period of time instead of making a point estimate by taking a single measurement. This is due to the fact that the point estimators only provide an approximate idea of the value of the parameter to be estimated, without knowing how good the approximation is. For this purpose, measurements at constant distances and concentrations were made by varying the number of values per measurement taken into account for each sensor instead of recording single values.

Aim: Reduction in oscillations at signals.

Procedure: First, we took a constant volume of 50 mL from the volatile substance at 100% concentration at a fixed distance of 2.5 cm. The number of recorded samples were: 10, 20, 30, 40, 60, 80 and 100 samples per measurement. These samples were recorded based on the Arduino sample execution time, and the mean readings were then reported.

#### 2.3.2. Step 2: Distance Influence

If a non-intrusive technique is used to analyze the gases emitted by bacteria, it is important to determine the distance range in which the sensors work properly as the volatile compounds emitted may disseminate in the environment. In order to achieve this, a systematic measurement was carried out where the distance range was evaluated within a uniform interval measured in terms of time. All experiments were independent and we changed the measurement distance as we cannot guarantee the distances in real wound assessment.

Aim: Determine the distance at which the sensors start losing sensitivity.

Procedure: Constant concentration of 50 mL of the volatile substance at 100% concentration was used, varying the distance from 2.5 cm to 13.3 cm with increments of 0.60 cm at every step.

#### 2.3.3. Step 3: Concentration Influence

In this experiment, the responses of the sensors at different concentrations were analyzed using a mixture of water and the volatile substance analyzed in liquid stage. If the concentrations of the gases emitted by *P. aeruginosa* and other skin and soft tissue pathogens are very low, it is important to ascertain the sensors’ sensitivity and range of accuracy.

Aim: Evaluate sensors’ sensitivity at different concentrations.

Procedure: The distance to perform the measurements was fixed at 2.5 cm, and concentration of the water-sample mixture was 50 mL with 5%, 20%, 50%, 80% and 100% of the volatile substance ammonia, alcohol and acetone analyzed and dry ice for the CO_2_.

### 2.4. System Design

In order to detect different types of gas emitted by the bacteria presented, we used one specific gas sensor for each gas. These gas sensors are connected to an Arduino Mega 2560 board which reads their values, and these values are then sent by USB to a Raspberry Pi board which displays and saves the data on a touch screen (Figure 2).

The hardware shown in Figure 2 provides real-time data collection and allows access with a friendly guided user interface. Additionally, the data is sent by WiFi to the hospital where the patient is located.

## 3. Results and Discussion

In this section, the results obtained by the selected gas sensors are presented. Firstly, the results in the validation process will be presented and secondly, the measurements with the six *P. aeruginosa* strains will be discussed, followed by the other six species.

### 3.1. Step 1: Operational Sample Rate

The steps in this experiment will be shown in detail by way of the TGS-826 sensor in ammonia detection. In Figure 3 the box-plots of the oscillation ranges are provided for each sample. It can be observed that for lower ppm range both 10 and 30 samples evidenced lower variability, implying that their signals will fluctuate less.

The standard deviation was calculated in each case to confirm the variability of each sample rate, and the evolution of this experiment over time is represented in Figure 4.

In Figure 4 the greatest oscillations in measurements were present in the cases where there were 20 and 100 samples per measurement, with differences of around 25 ppm. The lowest standard deviation was found in the case with 30 samples per measurement with σ = 3.7 ppm. Therefore, an average 30 samples per measurement was to be used for the experiments.

A summary of the oscillation rates and standard deviations for all four gas sensors is shown in Table 5.

### 3.2. Step 2: Distance Influence

The evolution of the ammonia concentration in parts per million (ppm) as a function of each distance over time is shown in Figure 5.

The signals in Figure 5 do not illustrate major oscillations and reveal a clear dependence on distance. For distances greater than about 4 cm between the source and the sensor, the change in concentration is negligible. Therefore, measurements should be made at distances of less than 4 cm in this case.

A summary of the distances at which the sensors lose their sensitivity under their pure substances is shown in Table 6.

### 3.3. Step 3: Concentration Influence

Concentration of the four gases detected by the sensors as a function of time for each solution is shown in Figure 6. Figure 6 shows an increase in the response of all sensors when increasing the concentration of the gases. It was noted that the sensors varied considerably for small concentrations. Nevertheless, for values higher than 80%, the responses of the sensors did not indicate any evident variations due to the internal resistance of the sensor which behaves in a logarithmic way, losing sensitivity until the saturation value is reached.

### 3.4. P. aeruginosa

In this subsection the results obtained with real bacteria are analyzed using the operational sample rates calculated in Table 5. The results obtained in detecting gases emitted by *P. aeruginosa* bacteria through gas sensors have been analyzed at different concentrations of colony-forming units (CFU) in two different mediums: solid and liquid. Estimated bacterial concentrations were approximately 10^8^ CFU/mL for liquid cultures and 10^8^ CFU/cm^2^ for solid cultures. The measuring procedure carried out should be the same in each sensor, measuring 2.5 mm from the samples with increments of 2.5 mm up to when the sensors lose their sensitivity.

In Figure 7 and Figure 8, the normalized responses of the four gas sensors are shown as being between 0 and 100 (y axis) and their evolution in time (x axis) through four colored lines (blue for ammonia, yellow for acetone, green for CO_2_ and red for alcohol). These responses are made up of two types of line for each sensor: the solid lines correspond to the measurement detected by the sensor in response to the bacteria and the dashed lines correspond to the base values that were recorded in situ in the laboratory without the presence of any bacteria (background noise which acts as a negative control for this experiment).

Both figures show that the four gas sensors recorded a response above the background signals in all cultures analyzed at 2.5 mm with concentrations of 10^8^ CFU/cm^2^ and 10^8^ CFU/mL. This would indicate that the sensors selected are capable of detecting the gases emitted by the *P. aeruginosa*.

Also, we observed that the response lines of the sensors approach the baselines as the sensor-strain distance increases, with sensitivity of the gases reaching 5 mm in all samples. This could be due to the fact that the amount of reagent that the samples had was not enough for the sensors to detect them at greater distances. Compared to the results under pure substances presented in Table 6, it can be observed that the distances at which the sensors lose their sensitivity descend from 4 cm to 5 mm due to the fact that bacteria emit a lower intensity of gases than the gases emitted by pure substances.

Ammonia: the response in the ammonia sensor recorded the highest values for the experimental samples over the background signals with a difference between values of up to 60 units in the solid samples and 30 units in the liquid samples. This is much greater than the differences observed with the other three gas sensors. For solid cultures, this difference was maintained at both 2.5 and 5 mm heights; however, in liquid cultures, this vast difference between sample and background was only observed in the case of the 2.5 mm height but not the 5 mm height.

Acetone: when observing the yellow lines, a major variation was noted over time at the 2.5 mm height and between the responses of 2.5 mm and 5 mm in the solid samples. In the liquid cultures, the experimental samples showed a slightly higher reading than the background measurements. Therefore, this sensor would be less useful than the ammonia sensor in detecting *P. aeruginosa*.

CO_2_: the responses of the CO_2_ gas sensor did not vary from 2.5 mm to 5 mm in the solid samples. The differences between measurement values and the background signal are considered insignificant since they do not exceed 10 units. Therefore, this sensor was unable to detect the *P. aeruginosa* strain PAO1 analyzed.

Alcohol: despite recording some level of sample detection in the solid sample, the response was not appreciably higher than the background signal. In the liquid sample, we observed a weak signal over the background readings because the sample measurement lines varied very little in terms of the baselines. This difference is much smaller compared to what was observed in the ammonia sensor.

It is important to note that the greatest responses corresponded to the TGS-826 ammonia and amine sensor since the distances between baselines and values of the measurement lines were the greatest. The authors M. J. Anand and Dr. V. Sridhar already found in [29] detections of odors produced by *P. aeruginosa* present in contaminated milk, ending up recording high voltage responses using the gas sensor TGS-822 acetone and the TGS-826 ammonia gas sensor. Additionally, a previous study had already shown recorded responses in *P. aeruginosa* strains with gas sensors [30]. This study therefore follows the line pursued by said experiments and confirms that *P. aeruginosa* can be detected using gas sensors. The results of the six clinical strains of *P. aeruginosa* in solid and liquid cultures obtained by the selected sensors are described in Appendix B.

It is also noteworthy that the responses in a solid medium were generally slightly higher than the responses in a liquid medium, which could be due to the solid strains giving off a certain odor caused by the emission of various volatile substances that might have been captured by the sensors. This would seem to indicate that the sensors are capable of detecting the odors produced by these bacteria, which would mean they are able to detect the decomposition of wounds.

### 3.5. Other Types of Bacteria Related to Wounds

*P. aeruginosa* is not the only infectious agent in wound infections producing volatile gases. Therefore, we wanted to include in the analysis with the proposed system other bacteria usually known for produce infections on skin and soft tissue. In order to do so, we tested a variety of bacteria and fungi at a height of 5 mm with all four gas sensors. The mean values of the gas concentrations in ppm are shown in Figure 9.

These results show the differentiation in the emission of gases by different types of bacteria compared to *P. aeruginosa*. Most of the gases emitted recorded the highest values in *P. aeruginosa* samples, which confirms the importance of this experiment and their value in designing such types of sensors.

Ammonia: the ammonia sensor revealed high levels of detection in the genera of *Candida, Staphylococcus*, and *Pseudomonas*, but was barely detectable in 8 of the 19 microbes analyzed. In the six types of *S. aureus* analyzed, sensitivity was recorded only in strains TJB008, TJB010 and NRS101, demonstrating that the behavior in this species of bacteria is not consistent among all isolates. In the *P. aeruginosa* species, many more parts per million of ammonia and detectable amines were emitted, and it could therefore be considered that the gas sensors used, in particular the TGS-826 sensor of ammonia and amine, are a good *P. aeruginosa* detector, and detecting their values could be an indication of the presence of this species.

Acetone: the ppm levels of acetone were similar in the species analyzed although of lesser magnitude than for ammonia and amines. The gas detectable by the acetone sensor might indicate the presence of bacteria, although the species with which the wound might be infected could not be determined.

Alcohol: recorded ppm values of alcohol present in the samples evidenced low variability with little detection. In contrast, ammonia and amines measurements were always consistent and constant at least in the case of *P. aeruginosa*.

CO_2_: similar sensitivities and overall levels were detected in the 19 microbes analyzed; however, CO_2_ was ubiquitous in most environments and detection using this sensor could be conflated between environmental sources and CO_2_ produced by the microbes in the wounds.

Previous groups had already studied *E. coli* gas emissions using gas sensor arrays with the same level of concentration detection as that described in this paper (10^8^ CFU) [31]. Furthermore, another research team had already described experiments conducted to assess ammonia production by *E. coli* species present in contaminated water sources using the same TGS-826 sensor as the one used in this study [32]. In the previous study mentioned, *E. coli* produced a detectable voltage reading using the 3.5 V sensor with the same concentration of bacteria as used in our study. Thus, although in our case there was no detectable ammonia response in *E. coli*, its detection could not be ruled out using the TGS-826 sensor.

Figure 10 shows the responses of normalized sensors between 0 and 100. Normalization was carried out taking into account the values recorded by each sensor separately, dividing each value by the maximum.

Figure 10 shows a greater response in the alcohol sensor despite the fact that this sensor has a much lower detection scale than the rest of the sensors (from 0 to 10 ppm versus a scale from 0 to 300 ppm for ammonia, for example).

## 4. Conclusions and Future Work

In this paper we have presented the investigation results of selected gas sensors—TGS-826, MQ-3, MQ-135 and MQ-138—for real-time bacteria detection in pressure ulcers with a non-intrusive technique. A system formed by multiple sensors requires different preprocessing techniques for each sensor due to variability in sensitivity and operational parameters.

The four sensors used detected the ammonia, CO_2_, acetone, and alcohol gases emitted by different types of microbes typically found in wound infections. These sensors detected the greatest measurements at 2.5 mm in height and lost their sensitivity at a range of 7.5 mm in the measurements with bacteria, and at a range of 4 cm under pure substances detectable by the sensors without bacteria.

Among the species analyzed, the TGS-826 sensor detected that *Staphylococcus* emitted gas ammonia in four of the six samples measured: TJB008, TJB010, TJB007 and NRS101. On the other hand, this sensor recorded the highest and most consistent responses in the six *P. aeruginosa* strains. This suggests that the TGS-826 sensor is a good choice for a low-cost option to detect *P. aeruginosa*.

Regarding the gases acetone and CO_2_, both were emitted by nearly all the bacteria analyzed but with lower intensity than ammonia emission.

The two culture mediums used for bacterial growth revealed greater gas emission detected by the sensors in cultures grown on solid agar plates compared to those grown under liquid conditions. Since results are promising, our future steps include evaluation of these sensors under real wound conditions as soon as we can make appropriate arrangements.

## Figures and Tables

**Figure 1 sensors-19-01523-f001:**
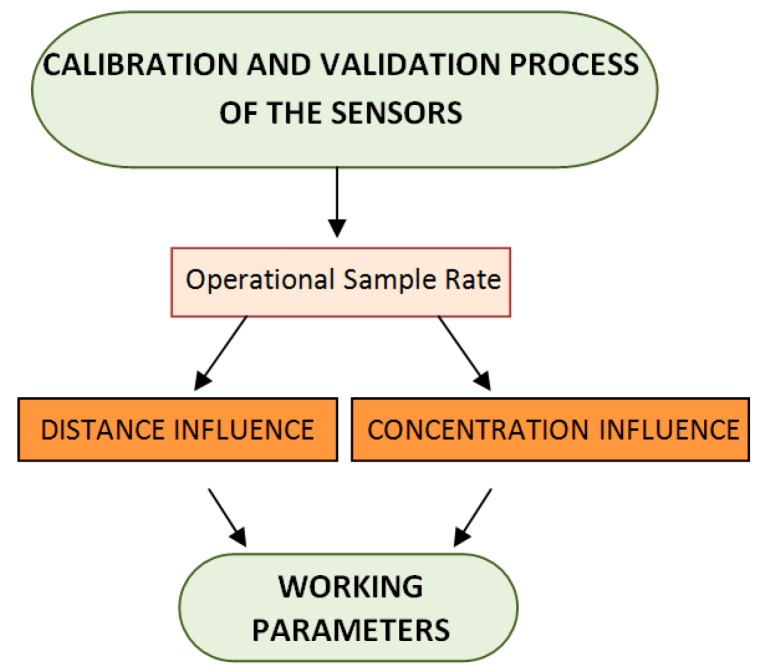
Work flow carried out for each sensor.

**Figure 2 sensors-19-01523-f002:**
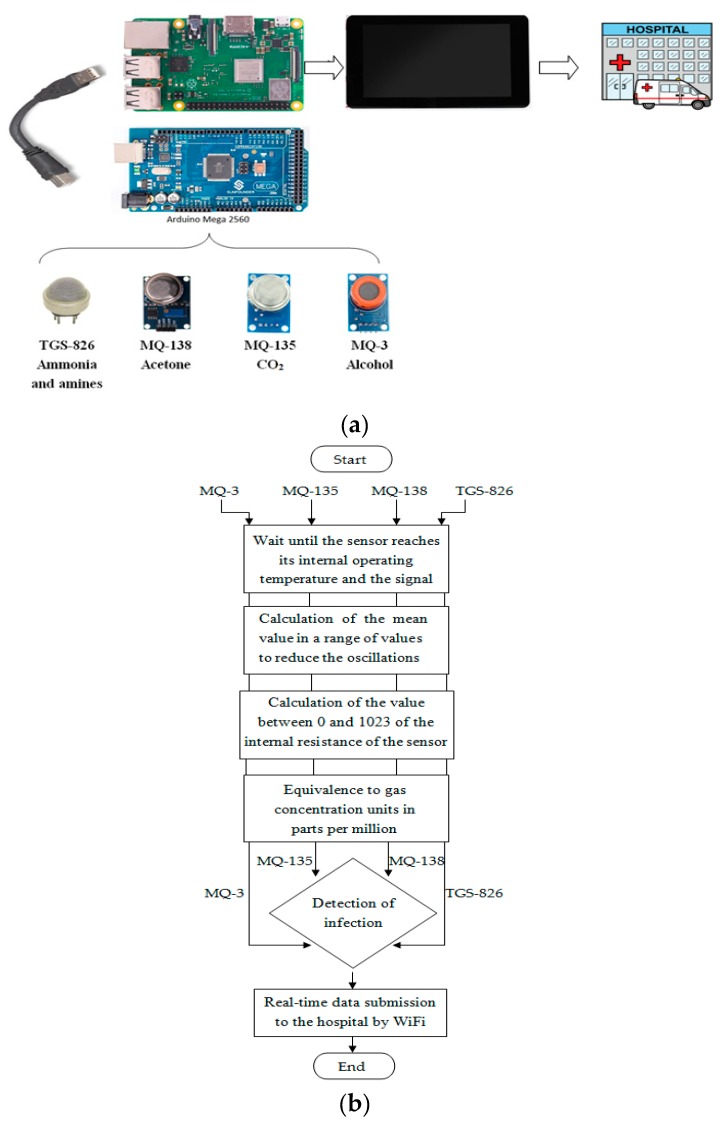
(**a**) system design; (**b**) high level schema for bacteria detection using four gas sensors.

**Figure 3 sensors-19-01523-f003:**
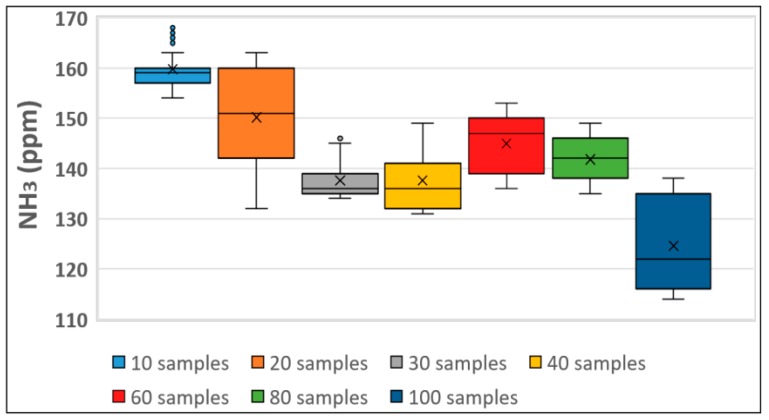
Oscillation range of TGS-826 detection which detects ammonia and amines.

**Figure 4 sensors-19-01523-f004:**
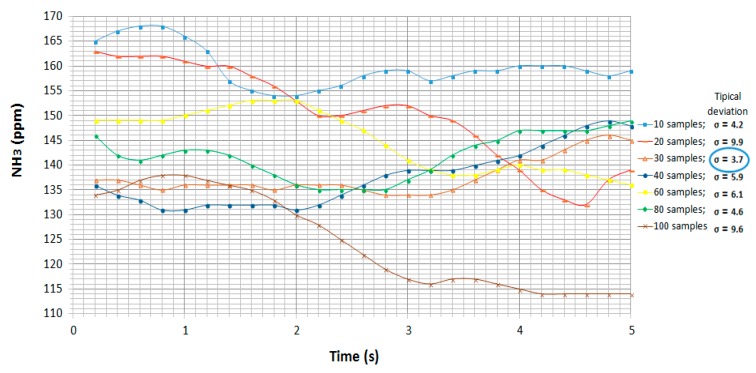
Evolution of NH_3_ and standard deviations over time using the TGS-826 sensor.

**Figure 5 sensors-19-01523-f005:**
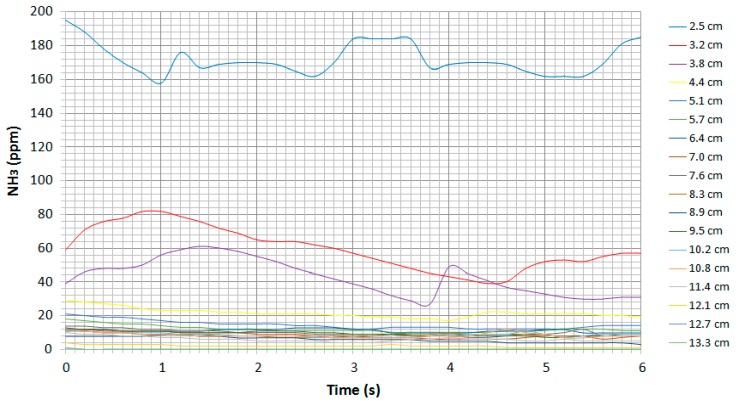
Evolution of ammonia ppm as a function of each distance using the TGS-826 sensor.

**Figure 6 sensors-19-01523-f006:**
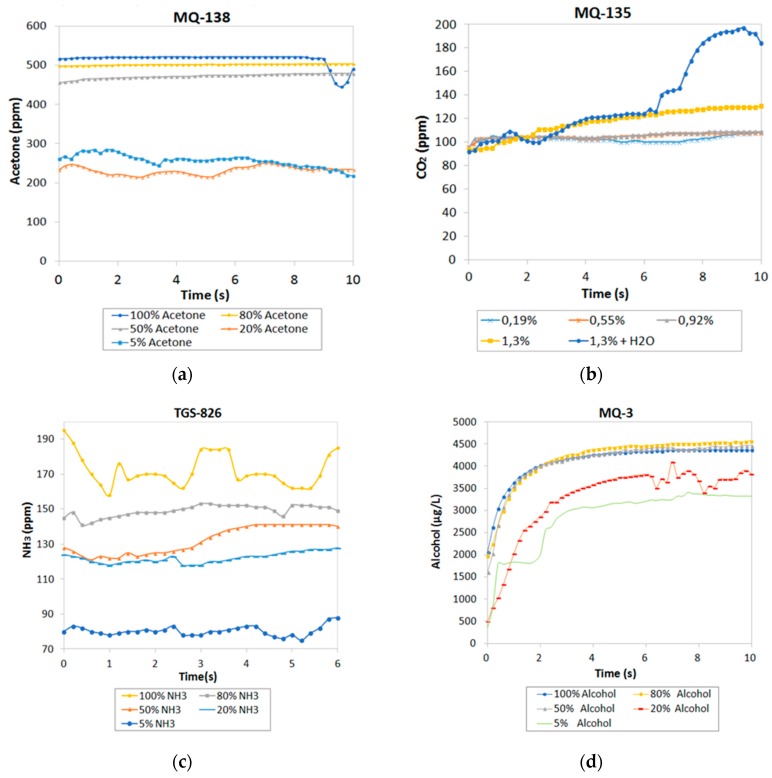
Evolution of gases at different concentrations: (**a**) acetone; (**b**) CO_2_; (**c**) NH_3_; (**d**) alcohol.

**Figure 7 sensors-19-01523-f007:**
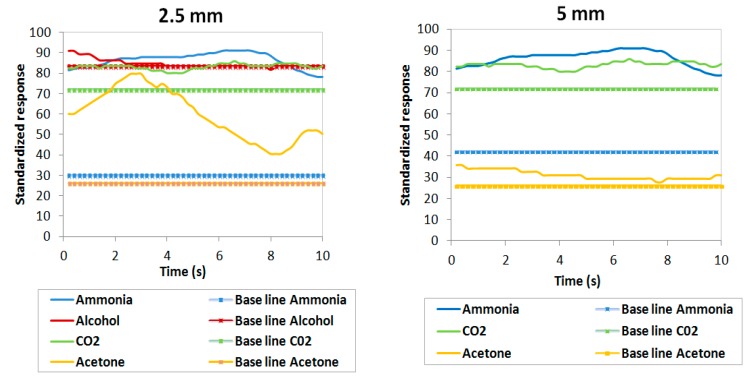
Responses of the four sensors in *P. aeruginosa* PAO1 in solid cultures with 10^8^ CFU/cm^2^.

**Figure 8 sensors-19-01523-f008:**
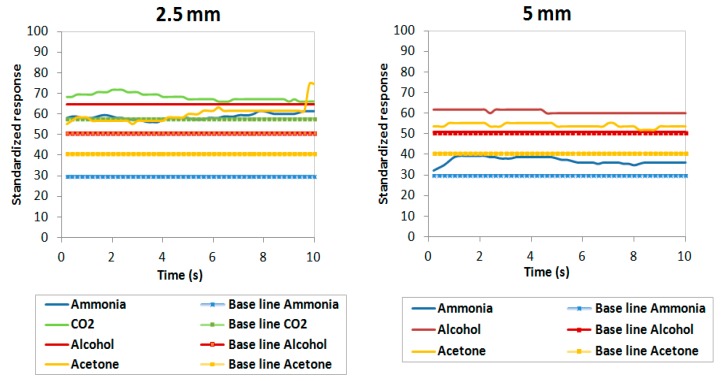
Responses of the four sensors in *P. aeruginosa* PAO1 in liquid cultures with 10^8^ CFU/mL.

**Figure 9 sensors-19-01523-f009:**
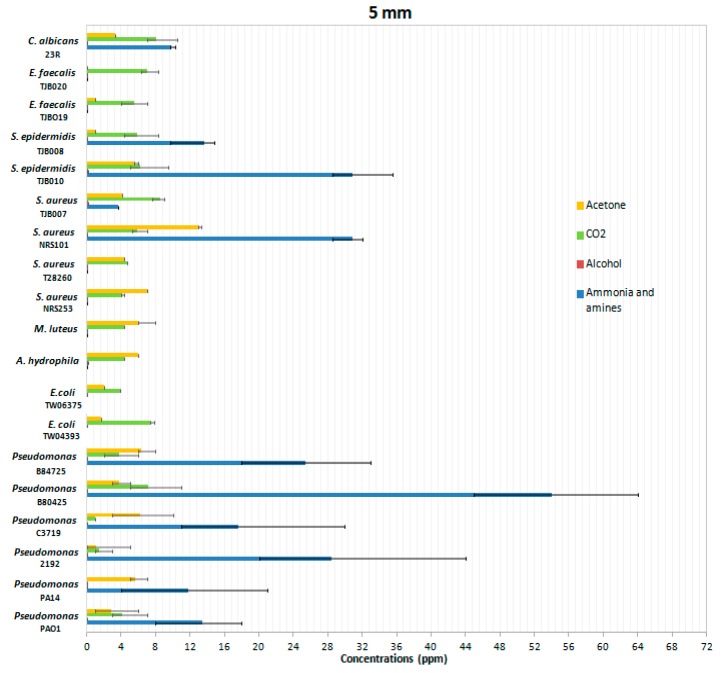
Mean gas concentrations produced by infectious microbial agents in solid cultures. Error bars represent one standard deviation of the data from at least three biological replicates.

**Figure 10 sensors-19-01523-f010:**
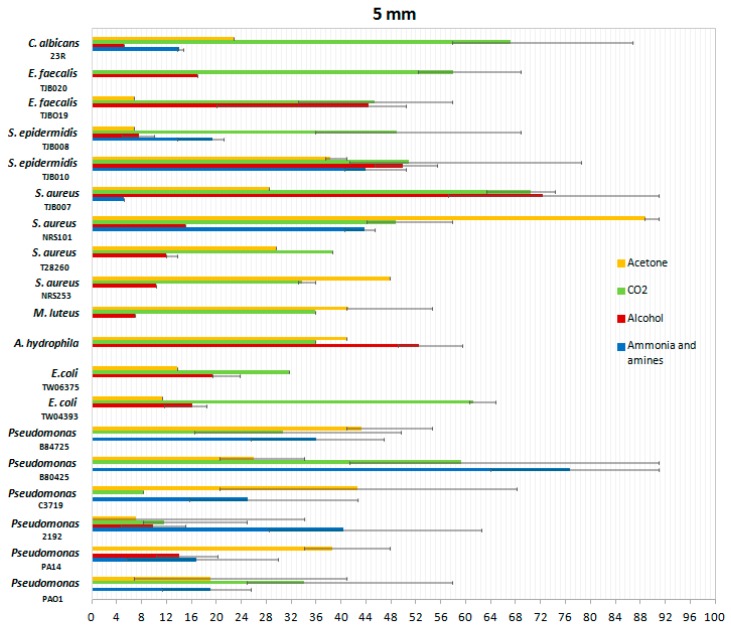
Responses normalized from 0 to 100 by each sensor separately, dividing each value by the maximum in all the bacteria.

**Table 1 sensors-19-01523-t001:** Inclusion criteria of the sensors selected.

Inclusion Criteria
All sensors used should be able to detect the specific type of gas emitted by the bacteria presented.
The detection scope should be suitable for internal or external spaces, ranging from 0 ppm to 1,000 ppm.
The sensor should be accurate to at least 20 ppm of volatile compounds in the air.
Small concentrations should be able to be detected, as concentration levels of gas emitted by the Pseudomonas is low.

**Table 2 sensors-19-01523-t002:** Exclusion criteria of the sensors selected.

Exclusion Criteria
Detection range equal to or above 1,000 ppm.
Accuracy below 20 ppm.
Voltage greater than 5 V, which would prevent its use in Arduino hardware and would require an external power source.

**Table 3 sensors-19-01523-t003:** Technical characteristics of the sensors selected.

Sensor	Chemicals Detected	Detection Scope	Accuracy (Maximum Value)	V_H_ (V)	A_H_ (mA)	R_H_ (Ω)	P_H_ (mW)
**TGS-826**	ammonia and amines	(30–300) ppm	± 12 ppm, 4%	5	< 167	30	< 833
**MQ-135**	CO_2_	(10–1,000) ppm	± 20 ppm, 2%	5	< 151.5	33	< 800
**MQ-3**	alcohol	(0.05–10) ppm	± 0.2 ppm, 2%	5	< 151.5	33	< 750
**MQ-138**	acetone	(5–500) ppm	± 20 ppm, 4%	5	< 170	31	< 850

**Table 4 sensors-19-01523-t004:** *P. aeruginosa* strain descriptions.

*P. aeruginosa* Strain	Description
**B80425**	Clinical cystic fibrosis isolate
**B84725**	Clinical cystic fibrosis isolate
**C3719**	Clinical cystic fibrosis isolate, small colony variant
**C2192**	Clinical cystic fibrosis isolate, mucoid variant
**PAO1**	Historic burn isolate, cultured in labs for more than 50 years
**PA14**	Historic burn isolate, cultured in labs for more than 50 years

**Table 5 sensors-19-01523-t005:** Operational sample rate and standard deviation for each sensor.

Sensor	Operational Sample Rate	Standard Deviation
**TGS-826** **Ammonia and amines**	30 samples per measurement	σ = 3.7 ppm
**MQ-135** **CO_2_**	20 samples per measurement	σ = 0.46 ppm
**MQ-3** **Alcohol**	10 samples per measurement	σ = 15.1 μg/L
**MQ-138** **Acetone**	10 samples per measurement	σ = 1.34 ppm

**Table 6 sensors-19-01523-t006:** Distances at which all four sensors lose their sensitivity under pure substances.

Sensor	Distance at Which the Sensor Loses Its Sensitivity Under Pure Substances
**TGS-826, Ammonia and amines**	4 cm
**MQ-135, CO_2_**	4 cm
**MQ-3, Alcohol**	7.6 cm
**MQ-138, Acetone**	4 cm

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
