# Peer review of "Detection of Volatile Compounds Emitted by Bacteria in Wounds Using Gas Sensors"

_sensors, 2019, doi:10.3390/s19071523_

Reviewer 1 Report

I do not understand the measurement protocols: sample number, concentrations, mode to introduce the gas or vapor in the sensor chamber, what is it measured as response, etc....

Author Response

Dear reviewer,

We are pleased to resubmit the revised version of the article titled Detection of Volatile Compounds Emitted by Bacteria in Wounds  using Gas Sensors .We really appreciate your constructive recommendations. We have addressed each of your concerns as outlined below.

Reviewer suggestions: I do not understand the measurement protocols: sample number, concentrations, mode to introduce the gas or vapor in the sensor chamber, what is it measured as response, etc….

Authors’ answer:

Thank you so much for your comments, we have sent the paper to a native English speaker in order to carry out an extensive English editing. Additionally, we made changes in the paper according to your suggestions:

Measurement protocol: we have gone through the protocol again to look for ways in which we could improve it. We have made some changes in the paper that are highlighted in the section 2.2 Experiment Design.

Sample number: we have increase in the paper the explanation about the sample numbers used in order to make the signals stable in the subsection 2.3.1 Operational Sample Rate. To achieve this, the read values that are recorded in the sensors, must correspond to an average of several values taken in a very short period of time instead of making a point estimate taking a single measurement. This is due to the fact that the point estimators only provide an approximate idea of the value of the parameter to be estimated, not knowing how good the approximation is.

Concentrations: to analyze the concentration influence in each sensor, we used pure substances in liquid stage at which the sensors register sensitivity. To change the gas intensity, we dilute these pure substances in water at different concentrations using a syringe.

Mode to introduce the gas in the sensor chamber: The factors that mainly affect the introduction of the gas in the sensor chamber are the adsorption attraction and the intermolecular force understood as a direct chemical interaction between the gas molecules and the semiconductor surface. This process works better under constant environment conditions (temperature, R.H. and pressure) [1].  When the gas is adsorbed, a charge transfer occurs between the gas and the semiconductor surface and this results in a change in surface conductivity [2]. At that moment, the sensor's conductivity increases depending on the gas concentration in the air and a simple electrical circuit can convert the change in conductivity to an output signal which corresponds to the gas concentration [3].

[1] D.  Hongfei; X. Guangzhong; S. Yuanjie; T. Huiling; D. Xiaosong; Y. He; Z. Qiuping, A New Model and Its Application for the Dynamic Response of RGO Resistive Gas Sensor. 2019, 19(4), 889. doi: https://doi.org/10.3390/s19040889.

[2] Gas sensors based on tin oxides. Available online: https://www.tdx.cat/bitstream/ handle/10803/2743/Tol1226.pdf

[3] Datasheet TGS-826 sensor. Available online: http://www.figarosensor.com /products/826Dtl.pdf.

What is it measured as response:  in the paper there are two different ways to represent the measurements with the bacteria. One way is the gas concentration registered in each case in ppm and the other way is the sensor response. The values of the responses correspond to a normalization between 0 and 100 taking into account all the gas concentration registered in each sensor separately in all the bacteria measured.

e.g. : If we measure 5 ppm of alcohol in one sample and the highest value that we recorded among all the samples and concentrations analyzed is 45 ppm, then the normalized response in that case would be:                                               

 (5 ppm/45 ppm) x 100 = 11,10.

We strongly appreciate your comments and, in order to enhance the paper, we have also improved the quality of some figures and we have reorganized the structure of the paper adding in appendix A and B some figures and tables. Additionally we have shorten the abstract and the conclusions in the paper as they were too long and we have done some format changes.

Reviewer 2 Report

This manuscript describes the detection of volatile gases emitted by bacteria in wounds using gas sensors. I think this manuscript is suitable for publication in this journal. My comments are as follows:

1- What is the sensing temperature? Did you test the effect of relative humidity in the sensing results? Did you test the stability of sensor arrays?

2- The first letter of all key words should be caps locked. Also full word for CFU should be written. Furthermore, you should be consistent in decimal numbers. 2.5 mm or 2,5 mm?

3- In introduction you should clearly define volatile species. What is definition of a volatile species?

4- What are the references for inclusion and exclusion criteria mentioned in Table 1?

5- It is better to remove words such as objective and protocol in each subsection of manuscript. It seems like a report rather than a scientific article.

6- Quality Fig. 5 is extremely poor. Figs 2-5 should be moved to Results and discussion section. Also Figs 7 and 8 can be moved to supporting information. In Fig. 9 & 10, there is no unit for y-axis. There are many tables in this manuscript. Some of them should be moved to supporting information.

7- Both Abstract and conclusion are too long and can be shorten.

Author Response

Comments to Reviewer

Detection of    Volatile Compounds Emitted by Bacteria in Wounds using Gas Sensors

Carlos    Salinas Alvarez , Daniel Sierra-Sosa     , Begonya Garcia-Zapirain  ,    Deborah Yoder-Himes  , Adel Elmaghraby

5/03/2019

Dear reviewer,

We are pleased to resubmit the revised version of the article titled Detection of Volatile Compounds Emitted by Bacteria in Wounds  using Gas Sensors .We really appreciate your constructive recommendations. We have addressed each of your concerns as outlined below.

Reviewer suggestion 1: What is the sensing temperature? Did you test the effect of humidity in the sensing results? Did you test the stability of sensor arrays?

Authors’ answer: Thank you so much for your suggestions, the tests were carried out in the biology laboratory of the co-author Dr. Deborah Yoder which is an isolate laboratory under establish environment conditions. In a second phase of the research, we are going to test the sensors with pressure ulcers in different patients. For this reason, during the measurements we verified that the environment conditions were similar than the conditions that the hospitals Basurto and Santa Marina in Bilbao, Spain have from which we already have an ethics committee approved to test with patients.

The measurements were performed under room temperature sensing between 20 °C and 22 °C, I have just included this data in the section 2, Methods. We are already working in that second phase of the project in which we will consider to vary the relative humidity to see the effect in the sensing results.

About the stability of sensor arrays, we took this into account because it is a very important point to be considered due to if the signals are not stable, the results could not have enough accuracy. For that, the read values that were recorded in the sensors, corresponded to an average of several values taken in a very short period of time instead of making a point estimate taking a single measurement. This is due to the fact that the point estimators only provide an approximate idea of the value of the parameter to be estimated, not knowing how good the approximation is. I have increased the explanation of this step in the 2.3.1 subsection (Operational Sample Rate).

Reviewer suggestion 2: The first letter of all the key words should be caps locked. Also full word for CFU should be written. Furthermore, you should be consistent in decimal numbers. 2.5 mm or 2,5 mm?

Authors’ answer: Thank you so much for your comment, we fully agree and I have already modified these changes in the paper. The full word for CFU in the subsection 3.4 P. aeruginosa.

Reviewer suggestion 3: In introduction you should clearly define volatile species. What is definition of a volatile species?

Authors’ answer: We appreciate your suggestion, certainly it should be clearly defined in the introduction to make sure that everybody understands it. A volatile gas is a substance that vaporizes readily from a liquid phase to a gas phase, We have added that definition in the introduction.

Reviewer suggestion 4 : What are the references for inclusion and exclusion criteria mentioned in Table 1?

Authors’ answer: I have added two references related to Table 1 in the subsection 2.1 Sensors selection.

Reviewer suggestion 5 : It is better to remove words such as objective and protocol in each subsection of manuscript. It seems like a report rather than a scientific article.

Authors’ answer: I have changed in the paper the words "objective" and "protocol" for other synonyms like aim, goal, procedure, follow steps, phases etc. Thank you so much for your advice.

Reviewer suggestion 6 : Quality Fig. 5 is extremely poor. Figs 2-5 should be moved to Results and discussion section. Also Figs 7 and 8 can be moved to supporting information. In Fig. 9 & 10, there is no unit for y-axis. There are many tables in this manuscript. Some of them should be moved to supporting information.

Authors’ answer: We strongly appreciate all your suggestions and comments. We have redone the Figure 5 and now the picture quality looks better and we have moved Figures 2-5 to Results and discussion section. We have also moved Figure 7 and 8 and the Tables 6-11 to the Appendix. Finally we have added y-axis to the Figure 9 and 10 saying that the values from 0 to 100 are Standardized responses (they have no units).

Reviewer suggestion 7: Both Abstract and conclusion are too long and can be shorten.

Authors’ answer: Thank you so much, we have shorten the abstract and the conclusions in the paper as they were too long.

Round  2

Reviewer 1 Report

After reading the answers and the new manuscript, I have many doubts yet.

It is difficult to relate the results obtained with artificial samples with real samples (bacteria). 

Why do not they measure amines?

When you have a wound there are very different bacteria at the same time, and they produce, many times, equal gases. How is it possible to diferenciate the type of bacteria? In this way, it is not possible. The resistive sensors are not selective. A classification and discrimination method should be used.

I do not understand how when the number of samples is greater the standar desviation is worse. This is not logical.

Why, for the other typed of bacteria, is the distance 5 mm instead 2.5 mm (this is the optimized).

It is not necessary to describe what is a volatile compound. this is known for everybody.

Author Response

Comments to Reviewer

Detection of    Volatile Compounds Emitted by Bacteria in Wounds Using Gas Sensors

Carlos    Salinas Alvarez , Daniel Sierra-Sosa     , Begonya Garcia-Zapirain  ,    Deborah Yoder-Himes  , Adel Elmaghraby

19/03/2019

Dear reviewer,

We are pleased to resubmit the revised version of the article titled Detection of Volatile Compounds Emitted by Bacteria in Wounds  Using Gas Sensors. We really appreciate your constructive recommendations. We have addressed each of your concerns as outlined below.

Thank you for your suggestion on the language: "English language and style are fine/minor spell check required", we have done some format corrections following the "English language guidelines for submissions to MDPI journals". Apart from that, it made us worried since we have hired a native speaking English proof reader in order to correct the grammar and style. w

We have also provided more background and included more references in the introduction and we have reread the conclusions in order to improve them and make them more supported by the results.

Here are our answers for your comments and suggestions:

Reviewer suggestion: It is difficult to relate the results obtained with artificial samples with real samples (bacteria).

Authors’ answer:

We are aware that in the current paper we did not include results of "real" wound assessment. Nonetheless, we created laboratory conditions to emulate the bacteria content in an infected wound, we are currently using the proposed system to take samples from real wounds with promising results. 

The value from this paper is that summarizes the feasibility of using low-cost resistive sensors to detect bacteria present when a wound is infected.  

Reviewer suggestion: Why do not they measure amines?

Authors’ answer:

We used TGS-826 sensor to measure amines, the results obtained for amines are described along the paper. Following your suggestion we included in appendix B the measurements of the different strains with all the used sensors.  

Reviewer suggestion: When you have a wound there are very different bacteria at the same time, and they produce, many times, equal gases. How is it possible to diferenciate the type of bacteria? In this way, it is not possible. The resistive sensors are not selective. A classification and discrimination method should be used.

Authors’ answer:

You are right, with the proposed system there is no way to differentiate which bacteria is producing the particular compounds, in session 3.5 we specify this issue and included the following statement:

P. aeruginosa is not the only infectious agent in wound infections producing volatile gases. Therefore, we wanted to include in the analysis with the proposed system other bacteria usually known for produce infections on skin and soft tissue.

Reviewer suggestion: I do not understand how when the number of samples is greater the standar desviation is worse. This is not logical.

Authors’ answer:

Each sensor has a specific working frequency, therefore, if many samples per measurement are taken in a really short time interval, the sensor may not have time to process all the measurements properly. If this happens, the data of each measurement can overlap with the rest and give rise to errors.

This is proof in the results we showed in the paper, seeing that each sensor registered lower oscillation and lower typical deviation depending on different samples per measurements. Our sensors did not register less fluctuations when we increased the samples per measurements in more than 30 samples per measurement.

Reviewer suggestion: Why, for the other typed of bacteria, is the distance 5 mm instead 2.5 mm (this is the optimized).

Authors’ answer:

All experiments were independent, as we cannot guarantee the distances in real wound assessment we changed the measurement distance, this comment is included in the subsection "2.3.2. Step 2: Distance influence".

Reviewer suggestion: It is not necessary to describe what is a volatile compound. this is known for everybody.

Authors’ answer:

We included this definition for generality. Nonetheless, following your suggestion we remove it.
